# ROBUSTNESS OF CLASSIFIERS TO UNIVERSAL PERTURBATIONS: A GEOMETRIC PERSPECTIVE

**Seyed Mohsen Moosavi Dezfooli**[*]
École Polytechnique Fédérale de Lausanne
`seyed.moosavi@epfl.ch`

**Alhussein Fawzi**[*†]
University of California, Los Angeles
`fawzi@cs.ucla.edu`

**Omar Fawzi**
École Normale Supérieure de Lyon
`omar.fawzi@ens-lyon.fr`

**Pascal Frossard**
École Polytechnique Fédérale de Lausanne
`pascal.frossard@epfl.ch`

**Stefano Soatto**
University of California, Los Angeles
`soatto@ucla.edu`

## ABSTRACT

Deep networks have recently been shown to be vulnerable to universal perturbations: there exist very small image-agnostic perturbations that cause most natural images to be misclassified by such classifiers. In this paper, we provide a quantitative analysis of the robustness of classifiers to universal perturbations, and draw a formal link between the robustness to universal perturbations, and the geometry of the decision boundary. Specifically, we establish theoretical bounds on the robustness of classifiers under two decision boundary models (*flat* and *curved* models). We show in particular that the robustness of deep networks to universal perturbations is driven by a key property of their curvature: there exist shared directions along which the decision boundary of deep networks is systematically positively curved. Under such conditions, we prove the existence of small universal perturbations. Our analysis further provides a novel geometric method for computing universal perturbations, in addition to explaining their properties.

## 1 INTRODUCTION

Despite the success of deep neural networks in solving complex visual tasks He et al. (2016); Krizhevsky et al. (2012), these classifiers have recently been shown to be highly vulnerable to perturbations in the input space. In Moosavi-Dezfooli et al. (2017), state-of-the-art classifiers are empirically shown to be vulnerable to universal perturbations: there exist very small *image-agnostic* perturbations that cause most natural images to be misclassified. The existence of universal perturbation is further shown in Hendrik Metzen et al. (2017) to extend to other visual tasks, such as semantic segmentation. Universal perturbations fundamentally differ from the random noise regime, and exploit essential properties of deep networks to misclassify most natural images with perturbations of very small magnitude. Why are state-of-the-art classifiers highly vulnerable to these specific directions in the input space? What do these directions represent? To answer these questions, we follow a theoretical approach and find the causes of this vulnerability in the geometry of the decision boundaries induced by deep neural networks. For deep networks, we show that the key to answering these questions lies in the existence of shared directions (across different datapoints) along which the decision boundary is highly curved. This establishes fundamental connections between geometry and robustness to universal perturbations, and thereby reveals new properties of the decision boundaries induced by deep networks.

---

[*]The first two authors contributed equally to this work.
[†]Now at Google DeepMind.

Our aim here is to derive an analysis of the vulnerability to universal perturbations in terms of the geometric properties of the boundary. To this end, we introduce two decision boundary models: 1) the *locally flat* model assumes that the first order linear approximation of the decision boundary holds locally in the vicinity of the natural images, and 2) the *locally curved* model provides a second order local description of the decision boundary, and takes into account the curvature information. We summarize our contributions as follows:

- Under the *locally flat* decision boundary model, we show that classifiers are vulnerable to universal directions as long as the normals to the decision boundaries in the vicinity of natural images are correlated (i.e., they approximately span a low dimensional space). This result formalizes and proves some of the empirical observations made in Moosavi-Dezfooli et al. (2017).

- Under the locally curved decision boundary model, the robustness to universal perturbations is instead driven by the *curvature* of the decision boundary; we show that the existence of *shared* directions along which the decision boundary is positively[1] curved implies the existence of very small universal perturbations.

- We show that state-of-the-art deep nets remarkably satisfy the assumption of our theorem derived for the locally curved model: there actually exist shared directions along which the decision boundary of deep neural networks are positively curved. Our theoretical result consequently captures the large vulnerability of state-of-the-art deep networks to universal perturbations.

- We finally show that the developed theoretical framework provides a novel (geometric) method for computing universal perturbations, and further explains some of the properties observed in Moosavi-Dezfooli et al. (2017) (e.g., diversity, transferability) regarding the robustness to universal perturbations.

## 2 DEFINITIONS AND NOTATIONS

Consider an $L$-class classifier $f : \mathbb{R}^d \to \mathbb{R}^L$. Given a datapoint $\boldsymbol{x} \in \mathbb{R}^d$, we define the estimated label $\hat{k}(\boldsymbol{x}) = \text{argmax}_k f_k(\boldsymbol{x})$, where $f_k(\boldsymbol{x})$ is the $k$th component of $f(\boldsymbol{x})$ that corresponds to the $k^{\text{th}}$ class. We define by $\mu$ a distribution over natural images in $\mathbb{R}^d$. The main focus of this paper is to analyze the robustness of classifiers to *universal* (image-agnostic) noise. Specifically, we define $\boldsymbol{v}$ to be a *universal* noise vector if $\hat{k}(\boldsymbol{x} + \boldsymbol{v}) \neq \hat{k}(\boldsymbol{x})$ for "most" $\boldsymbol{x} \sim \mu$. Formally, a perturbation $\boldsymbol{v}$ is $(\xi, \delta)$-universal, if the following two constraints are satisfied:

$$\|\boldsymbol{v}\|_2 \leq \xi,$$
$$\mathbb{P}\left(\hat{k}(\boldsymbol{x} + \boldsymbol{v}) \neq \hat{k}(\boldsymbol{x})\right) \geq 1 - \delta.$$

This perturbation image $\boldsymbol{v}$ is coined "universal", as it represents a fixed image-agnostic perturbation that causes label change for a large fraction of images sampled from the data distribution $\mu$. In Moosavi-Dezfooli et al. (2017), state-of-the-art classifiers have been shown to be surprisingly vulnerable to this simple perturbation regime.

It should be noted that universal perturbations are different from adversarial perturbations Szegedy et al. (2014); Biggio et al. (2013), which are datapoint-specific perturbations that are sought to fool a *specific* image. An adversarial perturbation is a solution to the following optimization problem

$$\boldsymbol{r}(\boldsymbol{x}) = \arg\min_{\boldsymbol{r} \in \mathbb{R}^d} \|\boldsymbol{r}\|_2 \text{ subject to } \hat{k}(\boldsymbol{x} + \boldsymbol{r}) \neq \hat{k}(\boldsymbol{x}), \tag{1}$$

which corresponds to the smallest additive perturbation that is necessary to change the label of the classifier $\hat{k}$ for $\boldsymbol{x}$. From a geometric perspective, $\boldsymbol{r}(\boldsymbol{x})$ quantifies the distance from $\boldsymbol{x}$ to the decision boundary (see Fig. 1a). In addition, due to the optimality conditions of Eq. (1), $\boldsymbol{r}(\boldsymbol{x})$ is orthogonal to the decision boundary at $\boldsymbol{x} + \boldsymbol{r}(\boldsymbol{x})$, as illustrated in Fig. 1a.

In the remainder of the paper, we analyze the robustness of classifiers to universal noise, with respect to the geometry of the *decision boundary* of the classifier $f$. Formally, the pairwise decision boundary,

---

[1]Throughout the paper, the sign of the curvature is chosen according to the normal vector, and the data point $x$, as illustrated in Fig. 3

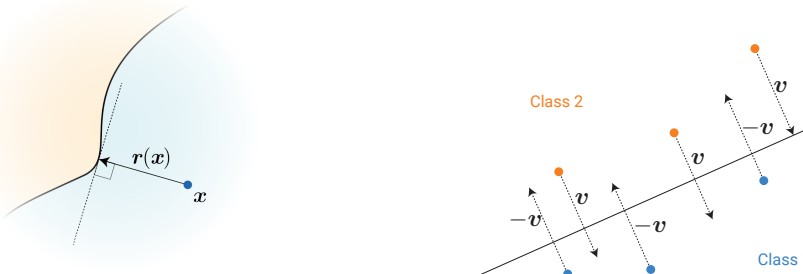

(a) Local geometry of the decision boundary.  (b) Universal direction $\boldsymbol{v}$ of a linear binary classifier.

Figure 1

when restricting the classifier to class $i$ and $j$ is defined by $\mathscr{B} = \{\boldsymbol{z} \in \mathbb{R}^d : f_i(\boldsymbol{z}) - f_j(\boldsymbol{z}) = 0\}$ (we omit the dependence of $\mathscr{B}$ on $i, j$ for simplicity). The decision boundary of the classifier hence corresponds to points in the input space that are equally likely to be classified as $i$ or $j$.

In the following sections, we introduce two models on the decision boundary, and quantify in each case the robustness of such classifiers to universal perturbations. We then show that the *locally curved* model better explains the vulnerability of deep networks to such perturbations.

## 3 Robustness of classifiers with flat decision boundaries

We start here our analysis by assuming a locally flat decision boundary model, and analyze the robustness of classifiers to universal perturbations under this decision boundary model. We specifically study the existence of a universal direction $\boldsymbol{v}$, such that

$$\hat{k}(\boldsymbol{x} + \boldsymbol{v}) \neq \hat{k}(\boldsymbol{x}) \text{ or } \hat{k}(\boldsymbol{x} - \boldsymbol{v}) \neq \hat{k}(\boldsymbol{x}), \tag{2}$$

where $\boldsymbol{v}$ is a vector of sufficiently small norm. It should be noted that a universal *direction* (as opposed to a universal vector) is sought in Eq. (2), as this definition is more adapted to the analysis of classifiers with locally flat decision boundaries. For example, while a binary linear classifier has a universal direction that fools all the data points, only half of the data points can be fooled with a universal vector (provided the classes are balanced) (see Fig. 1b). We therefore consider this slightly modified definition in the remainder of this section.

We start our analysis by introducing our local decision boundary model. For $\boldsymbol{x} \in \mathbb{R}^d$, note that $\boldsymbol{x} + \boldsymbol{r}(\boldsymbol{x})$ belongs to the decision boundary and $\boldsymbol{r}(\boldsymbol{x})$ is normal to the decision boundary at $\boldsymbol{x} + \boldsymbol{r}(\boldsymbol{x})$ (see Fig. 1a). A linear approximation of the decision boundary of the classifier at $\boldsymbol{x} + \boldsymbol{r}(\boldsymbol{x})$ is therefore given by $\boldsymbol{x} + \{\boldsymbol{v} : \boldsymbol{r}(\boldsymbol{x})^T \boldsymbol{v} = \|\boldsymbol{r}(\boldsymbol{x})\|_2^2\}$. Under this approximation, the vector $\boldsymbol{r}(\boldsymbol{x})$ hence captures the local geometry of the decision boundary in the vicinity of datapoint $\boldsymbol{x}$. We assume a local decision boundary model in the vicinity of datapoints $\boldsymbol{x} \sim \mu$, where the local classification region of $\boldsymbol{x}$ occurs in the halfspace $\boldsymbol{r}(\boldsymbol{x})^T \boldsymbol{v} \leq \|\boldsymbol{r}(\boldsymbol{x})\|_2^2$. Equivalently, we assume that outside of this half-space, the classifier outputs a different label than $\hat{k}(\boldsymbol{x})$. However, since we are analyzing the robustness to universal *directions* (and not vectors), we consider the following condition, given by

$$\mathscr{L}_s(\boldsymbol{x}, \rho) : \forall \boldsymbol{v} \in B(\rho), |\boldsymbol{r}(\boldsymbol{x})^T \boldsymbol{v}| \geq \|\boldsymbol{r}(\boldsymbol{x})\|_2^2 \implies \hat{k}(\boldsymbol{x} + \boldsymbol{v}) \neq \hat{k}(\boldsymbol{x}) \text{ or } \hat{k}(\boldsymbol{x} - \boldsymbol{v}) \neq \hat{k}(\boldsymbol{x}). \tag{3}$$

where $B(\rho)$ is a ball of radius $\rho$ centered at $\boldsymbol{0}$. An illustration of this decision boundary model is provided in Fig. 2a. It should be noted that linear classifiers satisfy this decision boundary model, as their decision boundaries are globally flat. This *local* decision boundary model is however more general, as we do *not* assume that the decision boundary is linear, but rather that the classification region in the vicinity of $\boldsymbol{x}$ is included in $\boldsymbol{x} + \{\boldsymbol{v} : |\boldsymbol{r}(\boldsymbol{x})^T \boldsymbol{v}| \leq \|\boldsymbol{r}(\boldsymbol{x})\|_2^2\}$. Moreover, it should be noted that the model being assumed here is on the decision boundary of the classifier, and not an assumption on the classification function $f$.[2] Fig. 2a provides an example of nonlinear decision boundary that satisfies this model.

---

[2] The decision boundary $\mathscr{B}$ is the zero level set of the functions $f_i - f_j$. $f$ can be a highly nonlinear function of the inputs, even when the zero-level set $\mathscr{B}$ is locally flat in the vicinity of datapoints.

In all the theoretical results of this paper, we assume that $\|r(x)\|_2 = 1$, for all $x \sim \mu$, for simplicity of the exposition. The results can be extended in a straightforward way to the case where $\|r(x)\|_2$ takes different values for points sampled from $\mu$. The following result shows that classifiers following the locally flat decision boundary model are *not* robust to small universal perturbations, provided the normals to the decision boundary (in the vicinity of datapoints) approximately belong to a low dimensional subspace of dimension $m \ll d$.

**Theorem 1.** *Let $\xi \geq 0, \delta \geq 0$. Let $\mathcal{S}$ be an $m$ dimensional subspace such that $\|P_{\mathcal{S}} r(x)\|_2 \geq 1 - \xi$ for almost all $x \sim \mu$,, where $P_{\mathcal{S}}$ is the projection operator on the subspace. Assume moreover that $\mathcal{L}_s(x, \rho)$ holds for almost all $x \sim \mu$, with $\rho = \frac{\sqrt{em}}{\delta(1-\xi)}$. Then, there exists a universal noise vector $v$, such that $\|v\|_2 \leq \rho$ and $\underset{x \sim \mu}{\mathbb{P}} \left( \hat{k}(x + v) \neq \hat{k}(x) \text{ or } \hat{k}(x - v) \neq \hat{k}(x) \right) \geq 1 - \delta$.*

The proof can be found in supplementary material, and relies on the construction of a universal perturbation through randomly sampling from $\mathcal{S}$. The vulnerability of classifiers to universal perturbations can be attributed to the *shared* geometric properties of the classifier's decision boundary in the vicinity of different data points. In the above theorem, this shared geometric property across different data points is expressed in terms of the normal vectors $r(x)$. The main assumption of the above theorem is specifically that normal vectors $r(x)$ to the decision boundary in the neighborhood of data points approximately live in a subspace $\mathcal{S}$ of low dimension $m < d$. Under this assumption, the above result shows the existence of universal perturbations of $\ell_2$ norm of order $\sqrt{m}$. When $m \ll d$, Theorem 1 hence shows that very small (compared to random noise, which scales as $\sqrt{d}$ Fawzi et al. (2016)) universal perturbations misclassifying most data points can be found.

**Remark 1.** Theorem 1 can be readily applied to assess the robustness of multiclass linear classifiers to universal perturbations. In fact, when $f(x) = W^T x$, with $W = [w_1, \ldots, w_L]$, the normal vectors are equal to $w_i - w_j$, for $1 \leq i, j \leq L, i \neq j$. These normal vectors exactly span a subspace of dimension $L - 1$. Hence, by applying the result with $\xi = 0$, and $m = L - 1$, we obtain that linear classifiers are vulnerable to universal noise, with magnitude proportional to $\sqrt{L - 1}$. In typical problems, we have $L \ll d$, which leads to very small universal directions.

**Remark 2.** Theorem 1 provides a partial expalanation to the vulnerability of deep networks, provided a locally flat decision boundary is assumed. Evidence in favor of this assumption was given through visualization of randomly chosen cross-sections in Warde-Farley et al. (2016); Fawzi et al. (2016). In addition, normal vectors to the decision boundary of deep nets (near data points) have been observed to approximately span a subspace $\mathcal{S}$ of sufficiently small dimension in Moosavi-Dezfooli et al. (2017). However, unlike linear classifiers, the dimensionality of this subspace $m$ is typically larger than the the number of classes $L$, leading to large upper bounds on the norm of the universal noise, under the flat decision boundary model. This simplified model of the decision boundary hence fails to exhaustively explain the large vulnerability of state-of-the-art deep neural networks to universal perturbations.

We show in the next section that the second order information of the decision boundary contains crucial information (*curvature*) that captures the high vulnerability to universal perturbations.

## 4  ROBUSTNESS OF CLASSIFIERS WITH CURVED DECISION BOUNDARIES

We now consider a model of the decision boundary in the vicinity of the data points that allows to leverage the *curvature* of nonlinear classifiers. Under this decision boundary model, we study the existence of universal perturbations satisfying $\hat{k}(x + v) \neq \hat{k}(x)$ for most $x \sim \mu$.[3]

We start by establishing an informal link between curvature of the decision boundary and robustness to universal perturbations, that will be made clear later in this section. As illustrated in Fig. 3, the norm of the required perturbation to change the label of the classifier along a specific direction $v$ is smaller if the decision boundary is positively curved, than if the decision boundary is flat (or with negative curvature). It therefore appears from Fig. 3 that the existence of universal perturbations (when the decision boundary is curved) can be attributed to the existence of *common* directions where

---

[3]Unlike for classifiers with locally flat decision boundaries, we now consider the problem of finding a universal *vector* (as opposed to universal *direction*) that fools most of the data points. This corresponds to the notion of universal perturbations first highlighted in Moosavi-Dezfooli et al. (2017).

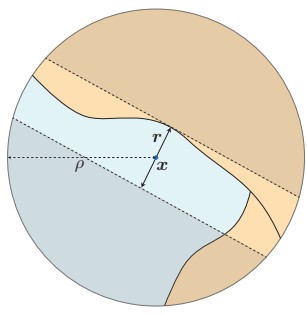
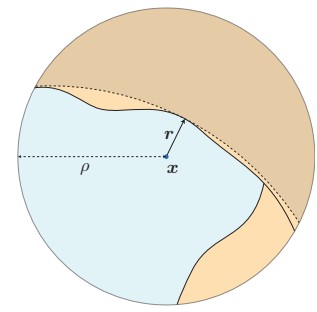

(a) Flat decision boundary model $\mathscr{L}_s(\boldsymbol{x}, \rho)$.       (b) Curved decision boundary model $\mathscr{Q}(\boldsymbol{x}, \rho)$.

Figure 2: Illustration of the decision boundary models considered in this paper. (a): For the flat decision boundary model, the set $\{\boldsymbol{v} : |\boldsymbol{r}(\boldsymbol{x})^T \boldsymbol{v}| \leq \|\boldsymbol{r}(\boldsymbol{x})\|_2^2\}$ is illustrated (stripe). Note that for $\boldsymbol{v}$ taken outside the stripe (i.e., in the grayed area), we have $\hat{k}(\boldsymbol{x} + \boldsymbol{v}) \neq \hat{k}(\boldsymbol{x})$ or $\hat{k}(\boldsymbol{x} - \boldsymbol{v}) \neq \hat{k}(\boldsymbol{x})$ in the $\rho$ neighborhood. (b): For the curved decision boundary model, the any vector $\boldsymbol{v}$ chosen in the grayed area is classified differently from $\hat{k}(\boldsymbol{x})$.

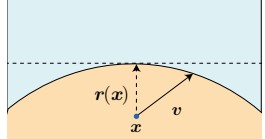
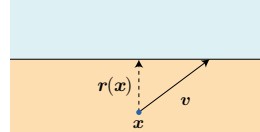
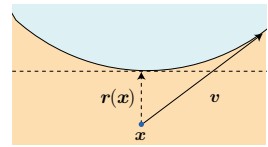

Figure 3: Link between robustness and curvature of the decision boundary. When the decision boundary is *positively* curved (left), small universal perturbations are more likely to fool the classifier.

the decision boundary is positively curved for many data points. In the remaining of this section, we formally prove the existence of universal perturbations, when there exists *common* positively curved directions of the decision boundary.

Recalling the definitions of Sec. 2, a quadratic approximation of the decision boundary at $\boldsymbol{z} = \boldsymbol{x} + \boldsymbol{r}(\boldsymbol{x})$ gives $\boldsymbol{x} + \{\boldsymbol{v} : (\boldsymbol{v} - \boldsymbol{r}(\boldsymbol{x}))^T H_{\boldsymbol{z}}(\boldsymbol{v} - \boldsymbol{r}(\boldsymbol{x})) + \alpha_x \boldsymbol{r}(\boldsymbol{x})^T(\boldsymbol{v} - \boldsymbol{r}(\boldsymbol{x})) = 0\}$, where $H_{\boldsymbol{z}}$ denotes the Hessian of $F$ at $\boldsymbol{z}$, and $\alpha_x = \frac{\|\nabla F(\boldsymbol{z})\|_2}{\|\boldsymbol{r}(\boldsymbol{x})\|_2}$, with $F = f_i - f_j$. In this model, the second order information (encoded in the Hessian matrix $H_{\boldsymbol{z}}$) captures the curvature of the decision boundary. We assume a *local* decision boundary model in the vicinity of datapoints $\boldsymbol{x} \sim \mu$, where the local classification region of $\boldsymbol{x}$ is bounded by a quadratic form. Formally, we assume that there exists $\rho > 0$ where the following condition holds for almost all $\boldsymbol{x} \sim \mu$:

$$\mathscr{Q}(\boldsymbol{x}, \rho) : \forall \boldsymbol{v} \in B(\rho), (\boldsymbol{v} - \boldsymbol{r}(\boldsymbol{x}))^T H_{\boldsymbol{z}}(\boldsymbol{v} - \boldsymbol{r}(\boldsymbol{x})) + \alpha_x \boldsymbol{r}(\boldsymbol{x})^T(\boldsymbol{v} - \boldsymbol{r}(\boldsymbol{x})) \leq 0 \implies \hat{k}(\boldsymbol{x} + \boldsymbol{v}) \neq \hat{k}(\boldsymbol{x}).$$

An illustration of this quadratic decision boundary model is shown in Fig. 2b. The following result shows the existence of universal perturbations, provided a subspace $\mathcal{S}$ exists where the decision boundary has positive curvature along most directions of $\mathcal{S}$:

**Theorem 2.** *Let $\kappa > 0, \delta > 0$ and $m \in \mathbb{N}$. Assume that the quadratic decision boundary model $\mathscr{Q}(\boldsymbol{x}, \rho)$ holds for almost all $\boldsymbol{x} \sim \mu$, with $\rho = \sqrt{\frac{2 \log(2/\delta)}{m}} \kappa^{-1} + \kappa^{-1/2}$. Let $\mathcal{S}$ be a $m$ dimensional subspace such that*

$$\mathbb{P}_{\boldsymbol{v} \sim \mathbb{S}} \left( \forall \boldsymbol{u} \in \mathbb{R}^2, \alpha_x^{-1} \boldsymbol{u}^T H_{\boldsymbol{z}}^{\boldsymbol{r}(\boldsymbol{x}), \boldsymbol{v}} \boldsymbol{u} \geq \kappa \|\boldsymbol{u}\|_2^2 \right) \geq 1 - \beta \text{ for almost all } \boldsymbol{x} \sim \mu,$$

*where $H_{\boldsymbol{z}}^{\boldsymbol{r}(\boldsymbol{x}), \boldsymbol{v}} = \Pi^T H_{\boldsymbol{z}} \Pi$ with $\Pi$ an orthonormal basis of $span(\boldsymbol{r}(\boldsymbol{x}), \boldsymbol{v})$, and $\mathbb{S}$ denotes the unit sphere in $\mathcal{S}$. Then, there is a universal perturbation vector $\boldsymbol{v}$ such that $\|\boldsymbol{v}\|_2 \leq \rho$ and $\mathbb{P}_{\boldsymbol{x} \sim \mu} \left( \hat{k}(\boldsymbol{x} + \boldsymbol{v}) \neq \hat{k}(\boldsymbol{x}) \right) \geq 1 - \delta - \beta$.*

The above theorem quantifies the robustness of classifiers to universal perturbations in terms of the curvature $\kappa$ of the decision boundary, along normal sections spanned by $\boldsymbol{r}(\boldsymbol{x})$, and vectors $\boldsymbol{v} \in \mathcal{S}$ (see

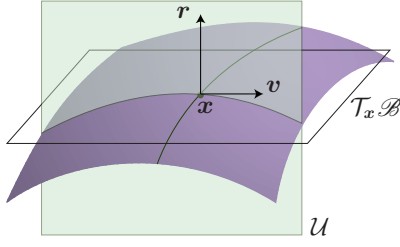 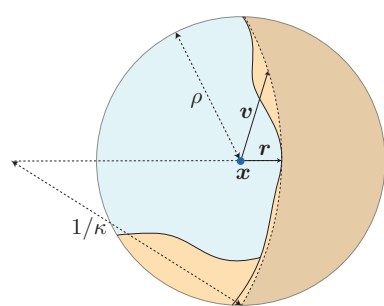

Figure 4: **Left:** Normal section $\mathcal{U}$ of the decision boundary, along the plane spanned by the normal vector $r(x)$ and $v$. **Right:** Geometric interpretation of the assumption in Theorem 2. Theorem 2 assumes that the decision boundary along normal sections $(r(x), v)$ is locally (in a $\rho$ neighborhood) located *inside* a disk of radius $1/\kappa$. Note the difference with respect to traditional notions of curvature, which express the curvature in terms of the osculating circle at $x + r(x)$. The assumption we use here is more "global".

Fig. 4 (*left*) for an illustration of a normal section). Fig. 4 (*right*) provides a geometric illustration of the assumption of Theorem 2. Provided a subspace $\mathcal{S}$ exists where the curvature of the decision boundary in the vicinity of datapoints $x$ is positive (along directions in $\mathcal{S}$), Theorem 2 shows that universal perturbations can be found with a norm of approximately $\frac{\kappa^{-1}}{\sqrt{m}} + \kappa^{-1/2}$. Hence, when the curvature $\kappa$ is sufficiently large, the existence of small universal perturbations is guaranteed with Theorem 2.[4]

**Remark 1.** We stress that Theorem 2 does *not* assume that the decision boundary is curved in the direction of all vectors in $\mathbb{R}^d$, but we rather assume the existence of a subspace $\mathcal{S}$ where the decision boundary is positively curved (in the vicinity of natural images $x$) along most directions in $\mathcal{S}$. Moreover, it should be noted that, unlike Theorem 1, where the normals to the decision boundary are assumed to belong to a low dimensional subspace, no assumption is imposed on the normal vectors. Instead, we assume the existence of a subspace $\mathcal{S}$ leading to positive curvature, for points on the decision boundary in the vicinity of natural images.

**Remark 2.** Theorem 2 does not only predict the vulnerability of classifiers, but it also provides a constructive way to find such universal perturbations. In fact, *random vectors* sampled from the subspace $\mathcal{S}$ are predicted to be universal perturbations (see supp. material for more details). In Section 5, we will show that this new construction works remarkably well for deep networks, as predicted by our analysis.

## 5 EXPERIMENTAL RESULTS: UNIVERSAL PERTURBATIONS FOR DEEP NETS

We first evaluate the validity of the assumption of Theorem 2 for deep neural networks, that is the existence of a low dimensional subspace where the decision boundary is positively curved along most directions sampled from the subspace. To construct the subspace, we find the directions that lead to large positive curvature in the vicinity of a given set of training points $\{x_1, \ldots, x_n\}$. We recall that principal directions $v_1, \ldots, v_{d-1}$ at a point $z$ on the decision boundary correspond to the eigenvectors (with nonzero eigenvalue) of the matrix $H_z^t$, given by $H_z^t = PH_zP$, where $P$ denotes the projection operator on the tangent to the decision boundary at $z$, and $H_z$ denotes the Hessian of the decision boundary function evaluated at $z$ Lee (2009). Common directions with large average curvature at $z_i = x_i + r(x_i)$ (where $r(x_i)$ is the minimal perturbation defined in Eq. (1)) hence correspond to the eigenvectors of the average Hessian matrix $\overline{H} = n^{-1} \sum_{i=1}^{n} H_{z_i}^t$. We therefore set our subspace, $\mathcal{S}_c$, to be the span of the first $m$ eigenvectors of $\overline{H}$, and show that the subspace constructed in this way satisfies the assumption of Theorem 2. To determine whether the decision boundary is positively curved in most directions of $\mathcal{S}_c$ (for unseen datapoints from the validation set), we compute the average curvature across random directions in $\mathcal{S}_c$ for points on the decision boundary,

---

[4]Theorem 2 should not be seen as a generalization of Theorem 1, as the models are distinct. In fact, while the latter shows the existence of universal *directions*, the former bounds the existence of universal *perturbations*.

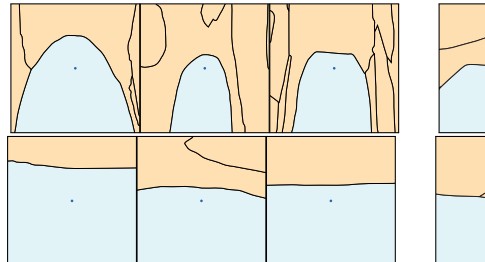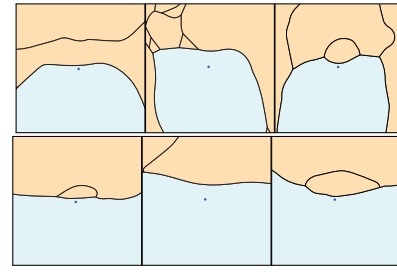

Figure 5: Visualization of normal cross-sections of the decision boundary, for CIFAR-10 (Left: LeNet, Right: ResNet-18). **Top:** Normal cross-sections along $(\boldsymbol{r}(\boldsymbol{x}), \boldsymbol{v})$, where $\boldsymbol{v}$ is the universal perturbation computed using the algorithm in Moosavi-Dezfooli et al. (2017). **Bottom:** Normal cross-sections along $(\boldsymbol{r}(\boldsymbol{x}), \boldsymbol{v})$, where $\boldsymbol{v}$ is a *random* vector uniformly sampled from the unit sphere in $\mathbb{R}^d$.

i.e. $\boldsymbol{z} = \boldsymbol{x} + \boldsymbol{r}(\boldsymbol{x})$; the average curvature is formally given by

$$\overline{\kappa}_{\mathcal{S}}(\boldsymbol{x}) = \mathop{\mathbb{E}}_{\boldsymbol{v} \sim \mathbb{S}} \left( \frac{(P\boldsymbol{v})^T H_{\boldsymbol{z}} (P\boldsymbol{v})}{\|P\boldsymbol{v}\|_2^2} \right),$$

where $\mathbb{S}$ denotes the unit sphere in $\mathcal{S}_c$. In Fig. 7 (a), the average of $\overline{\kappa}_{\mathcal{S}}(\boldsymbol{x})$ across points sampled from the *validation set* is shown (as well as the standard deviation) in function of the subspace dimension $m$, for a LeNet architecture LeCun et al. (1998) trained on the CIFAR-10 dataset.[5] Observe that when the dimension of the subspace is sufficiently small, the average curvature is strongly oriented towards positive curvature, which empirically shows the existence of this subspace $\mathcal{S}_c$ where the decision boundary is positively curved for most data points in the validation set. This empirical evidence hence suggests that the assumption of Theorem 2 is satisfied, and that universal perturbations hence represent random vectors sampled from this subspace $\mathcal{S}_c$.

To show this strong link between the vulnerability of universal perturbations and the *positive curvature* of the decision boundary, we now visualize normal sections of the decision boundary of deep networks trained on ImageNet (CaffeNet (Jia et al., 2014) and ResNet-152 (He et al., 2016)) and CIFAR-10 (LeNet (LeCun et al., 1998) and ResNet-18 (He et al., 2016)) in the direction of their respective universal perturbations.[6] Specifically, we visualize normal sections of the decision boundary in the plane $(\boldsymbol{r}(\boldsymbol{x}), \boldsymbol{v})$, where $\boldsymbol{v}$ is a universal perturbation computed using the universal perturbations algorithm of Moosavi-Dezfooli et al. (2017). The visualizations are shown in Fig. 5 and 6. Interestingly, the universal perturbations belong to highly positively curved directions of the decision boundary, despite the absence of any geometric constraint in the algorithm to compute universal perturbations. To fool most data points, universal perturbations hence naturally seek *common directions* of the embedding space, where the decision boundary is positively curved. These directions lead to very small universal perturbations, as highlighted by our analysis in Theorem 2. It should be noted that such *highly curved* directions of the decision boundary are rare, as random normal sections are comparatively flat (see Fig. 5 and 6, second row). This is due to the fact that most principal curvatures are approximately zero, for points sampled on the decision boundary in the vicinity of data points.

Recall that Theorem 2 suggests a novel procedure to generate universal perturbations; in fact, random perturbations from $\mathcal{S}_c$ are predicted to be universal perturbations. To assess the validity of this result, Fig. 7 (b) illustrates the fooling rate of the universal perturbations (for the LeNet network on CIFAR-10) sampled uniformly at random from the unit sphere in subspace $\mathcal{S}_c$, and scaled to have a fixed norm ($1/5$th of the norm of the random noise required to fool most data points). We assess the quality of such perturbation by further indicating in Fig. 7 (b) the fooling rate of the universal

---

[5]The LeNet architecture we used has two convolutional layers (filters of size 5) followed by three fully connected layers. We used SGD for training, with a step size 0.01 and a momentum term of 0.9 and weight decay of $10^{-4}$. The accuracy of the network on the test set is 78.4%.

[6]For the networks on ImageNet, we used the Caffe pre-trained models `https://github.com/BVLC/caffe/wiki/Model-Zoo`. The ResNet-18 architecture was trained on the CIFAR-10 task with stochastic gradient descent with momentum and weight decay regularization. It achieves an accuracy on the test of 94.18%.

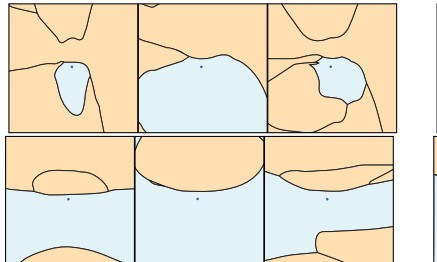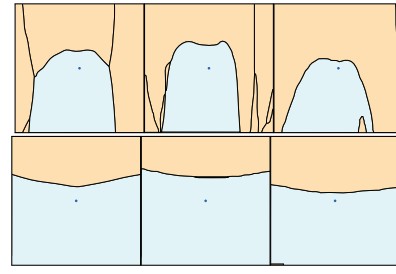

Figure 6: Visualization of normal cross-sections of the decision boundary, for ImageNet (Left: ResNet-152, and Right: CaffeNet) **Top:** Normal cross-sections along $(\boldsymbol{r}(\boldsymbol{x}), \boldsymbol{v})$, where $\boldsymbol{v}$ is the universal perturbation computed using the algorithm in Moosavi-Dezfooli et al. (2017). **Bottom:** Normal cross-sections along $(\boldsymbol{r}(\boldsymbol{x}), \boldsymbol{v})$, where $\boldsymbol{v}$ is a *random* vector uniformly sampled from the unit sphere in $\mathbb{R}^d$.

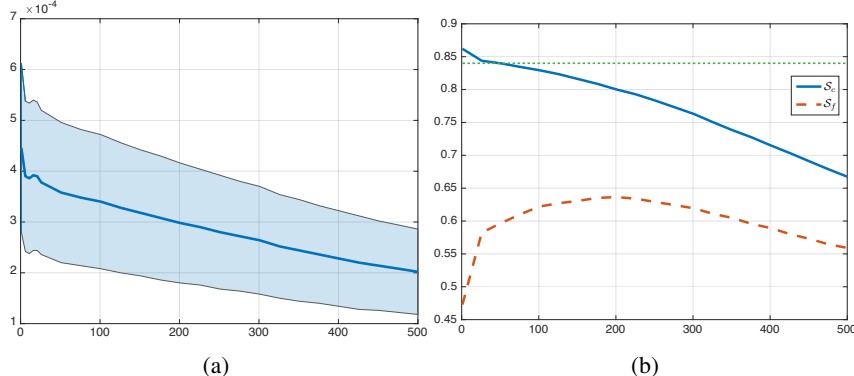

(a)                    (b)

Figure 7: **(a)** Average curvature $\overline{\kappa}_{\mathcal{S}}$, averaged over 1000 *validation* datapoints, as a function of the subspace dimension. **(b)** Fooling rate of universal perturbations (on an unseen *validation* set) computed using random perturbations in 1) $\mathcal{S}_c$: the subspace of positively curved directions, and 2) $\mathcal{S}_f$: the subspace collecting normal vectors $\boldsymbol{r}(\boldsymbol{x})$. The dotted line corresponds to the fooling rate using the algorithm in Moosavi-Dezfooli et al. (2017). $\mathcal{S}_f$ corresponds to the largest singular vectors corresponding to the matrix gathering the *normal vectors* $\boldsymbol{r}(\boldsymbol{x})$ in the training set (similar to the approach in Moosavi-Dezfooli et al. (2017)).

perturbation computed using the original algorithm in Moosavi-Dezfooli et al. (2017). Observe that random perturbations sampled from $\mathcal{S}_c$ (with $m$ small) provide very powerful universal perturbations, fooling nearly $85\%$ of data points from the validation set. This rate is comparable to that of the algorithm in Moosavi-Dezfooli et al. (2017), while using much less training points (only $n = 100$, while at least $2,000$ training points are required by Moosavi-Dezfooli et al. (2017)). The very large fooling rates achieved with such a simple procedure (random generation in $\mathcal{S}_c$) confirms that the curvature is the governing factor that controls the robustness of classifiers to universal perturbations, as analyzed in Section 4. In fact, such high fooling rates cannot be achieved by only using the model of Section 3 (neglecting the curvature information), as illustrated in Fig. 7 (b). Specifically, by generating random perturbations from the subspace $\mathcal{S}_f$ collecting normal vectors $\boldsymbol{r}(\boldsymbol{x})$ (which is the procedure that is suggested by Theorem 1 to compute universal perturbations, without taking into account second order information), the best universal perturbation achieves a fooling rate of $65\%$, which is significantly worse than if the curvature is used to craft the perturbation. We further perform in Appendix C the same experiment on other architectures (VGG-16 and ResNet-18) to verify the consistency of the results across networks. It can be seen that, similarly to Fig. 7 (b), the proposed approach of generating universal perturbations through random sampling from the subspace $\mathcal{S}_c$ achieves high fooling rates (comparable to the algorithm in Moosavi-Dezfooli et al. (2017), and significantly higher than by using $\mathcal{S}_f$).

Fig 8 illustrates a universal perturbation for ImageNet, corresponding to the maximally curved shared direction (or in other words, the maximum eigenvalue of $\overline{H}$ computed using $n = 200$ random samples).[7] The CaffeNet architecture is used, and Fig. 8 also represents sample perturbed images that fool the classifier. Just like the universal perturbation in Moosavi-Dezfooli et al. (2017), the perturbations are not very perceptible, and lead to misclassification of most unseen images in the validation set. For this example on ImageNet, the fooling rate of this perturbation is $67.2\%$ on the validation set. This is significantly larger than the fooling rate of the perturbation computed using $\mathcal{S}_f$ only ($38\%$), but lower than that of the original algorithm ($85.4\%$) proposed in (Moosavi-Dezfooli et al., 2017). We hypothesize that this gap for ImageNet is partially due to the small number of samples, which was made due to computational restrictions.

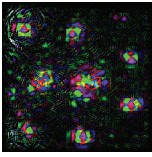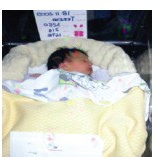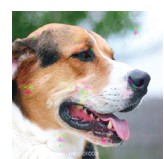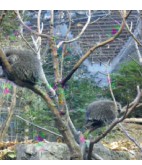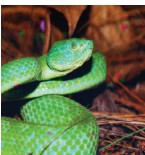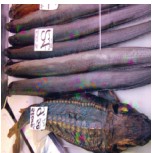

Figure 8: Left column: Universal perturbation computed through random sampling from $\mathcal{S}_c$. Second column to end: All images are (incorrectly) classified as "bubble". The CaffeNet architecture is used. Similarly to Moosavi-Dezfooli et al. (2017), the perturbation is constrained to have $\ell_2$ norm of $2,000$.

The existence of this subspace $\mathcal{S}_c$ (and that universal perturbations are random vectors in $\mathcal{S}_c$) further explains the high diversity of universal perturbations. Fig. 9 illustrates different universal perturbations for CIFAR-10 computed by sampling random directions from $\mathcal{S}_c$. The diversity of such perturbations justifies why re-training with perturbed images (as in Moosavi-Dezfooli et al. (2017)) does *not* significantly improve the robustness of such networks, as other directions in $\mathcal{S}_c$ can still lead to universal perturbations, even if the network becomes robust to some directions. Finally, it is interesting to note that this subspace $\mathcal{S}_c$ is likely to be shared not only across datapoints, but also different networks (to some extent). To support this claim, Fig. 10 shows the cosine of the principal angles between subspaces $\mathcal{S}_c^{\text{LeNet}}$ and $\mathcal{S}_c^{\text{NiN}}$, computed for LeNet and NiN Lin et al. (2014) models. Note that the first principal angles between the two subspaces are very small, leading to shared directions between the two subspaces. A similar observation is made for networks trained on ImageNet in the supp. material. The sharing of $\mathcal{S}_c$ across different networks explains the transferability of universal perturbations observed in Moosavi-Dezfooli et al. (2017).

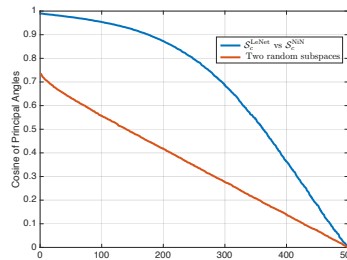

Figure 10: Cosine of principal angles between $\mathcal{S}_c^{\text{LeNet}}$ and $\mathcal{S}_c^{\text{NiN}}$. For comparison, cosine of angles between two random subspaces is also shown.

## 6   DISCUSSION AND RELATED WORK

In this paper, we analyzed the robustness of classifiers to universal perturbations, under two decision boundary models: Locally flat and curved. We showed that the first are not robust to universal directions, provided the normal vectors in the vicinity of natural images are correlated. While this model explains the vulnerability for e.g., linear classifiers, this model discards the curvature information, which is essential to fully analyze the robustness of deep nets to universal perturbations. The second, classifiers with *curved* decision boundaries, are instead not robust to universal perturbations, provided the existence of a shared subspace along which the decision boundary is positively curved (for most

---

[7]We used $m = 1$ in this experiment as the matrix $\overline{H}$ is prohibitively large for ImageNet.

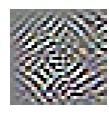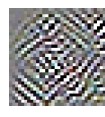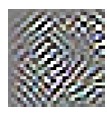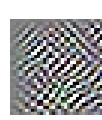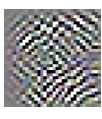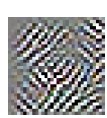

Figure 9: Diversity of universal perturbations randomly sampled from the subspace $\mathcal{S}_c$. The normalized inner product between two perturbations is less than $0.1$.

directions). We empirically verify this assumption for deep nets. Our analysis hence explains the existence of universal perturbations, and further provides a purely geometric approach for computing such perturbations, in addition to explaining properties of perturbations, such as their diversity.

Other authors have focused on the analysis of the robustness properties of SVM classifiers (e.g., Xu et al. (2009)) and new approaches for constructing robust classifiers (based on robust optimization) Caramanis et al. (2012); Lanckriet et al. (2003). More recently, some have assessed the robustness of deep neural networks to different regimes such as adversarial perturbations Szegedy et al. (2014); Biggio et al. (2013), random noise Fawzi et al. (2016), and occlusions Sharif et al. (2016); Evtimov et al. (2017). The robustness of classifiers to adversarial perturbations has been specifically studied in Szegedy et al. (2014); Goodfellow et al. (2015); Moosavi-Dezfooli et al. (2016); Carlini & Wagner (2017); Baluja & Fischer (2017), followed by works to improve the robustness Madry et al. (2017); Gu & Rigazio (2014); Papernot et al. (2015); Cisse et al. (2017), and attempts at explaining the phenomenon in Goodfellow et al. (2015); Fawzi et al. (2015); Tabacof & Valle (2016); Tanay & Griffin (2016). This paper however differs from these previous works as we study *universal (image-agnostic) perturbations* that can fool every image in a dataset, as opposed to image-specific adversarial perturbations that are not universal across datapoints (as shown in Moosavi-Dezfooli et al. (2017)). Moreover, explanations that hinge on the output of a deep network being well approximated by a linear function of the inputs $f(\boldsymbol{x}) = W\boldsymbol{x} + b$ are inconclusive, as the assumption is violated even for relatively small networks. We show here that it is precisely the large curvature of the decision boundary that causes vulnerability to universal perturbations. Our bounds indeed show an *increasing* vulnerability with respect to the curvature of the decision boundary, and represent up to our knowledge the first quantitative result showing tight links between robustness and curvature. In addition, we show empirically that the first-order approximation of the decision boundary is not sufficient to explain the high vulnerability to universal perturbations (Fig. 7 (b)). Recent works have further proposed new methods for computing universal perturbations Mopuri et al. (2017); Khrulkov & Oseledets (2017); instead, we focus here on an analysis of the phenomenon of vulnerability to universal perturbations, while also providing a constructive approach to compute universal perturbations leveraging our curvature analysis. Finally, it should be noted that recent works have studied properties of deep networks from a geometric perspective (such as their expressivity Poole et al. (2016); Montufar et al. (2014)); our focus is different in this paper as we analyze the robustness with the geometry of the decision boundary.

Our analysis hence shows that to construct classifiers that are robust to universal perturbations, it is key to *suppress* this subspace of shared positive directions, which can possibly be done through regularization of the objective function. This will be the subject of future works.

# A    PROOF OF THEOREM 1

We first start by recalling a result from Fawzi et al. (2016), which is based on Dasgupta & Gupta (2003).

**Lemma 1.** *Let $\boldsymbol{v}$ be a random vector uniformly drawn from the unit sphere $\mathbb{S}^{d-1}$, and $\mathbf{P}_m$ be the projection matrix onto the first $m$ coordinates. Then,*

$$\mathbb{P}\left(\beta_1(\delta, m)\frac{m}{d} \leq \|\mathbf{P}_m \boldsymbol{v}\|_2^2 \leq \beta_2(\delta, m)\frac{m}{d}\right) \geq 1 - 2\delta, \tag{4}$$

*with $\beta_1(\delta, m) = \max((1/e)\delta^{2/m}, 1 - \sqrt{2(1 - \delta^{2/m})})$, and $\beta_2(\delta, m) = 1 + 2\sqrt{\frac{\ln(1/\delta)}{m}} + \frac{2\ln(1/\delta)}{m}$.*

We use the above lemma to prove our result, which we recall as follows:

**Theorem 1.** *Let $\xi \geq 0, \delta \geq 0$. Let $\mathcal{S}$ be an $m$ dimensional subspace such that $\|P_{\mathcal{S}}\boldsymbol{r}(\boldsymbol{x})\|_2 \geq 1 - \xi$ for almost all $\boldsymbol{x} \sim \mu$,, where $P_{\mathcal{S}}$ is the projection operator on the subspace. Assume moreover that $\mathscr{L}_{\mathrm{s}}(\boldsymbol{x}, \rho)$ holds for almost all $\boldsymbol{x} \sim \mu$, with $\rho = \frac{\sqrt{e}m}{\delta(1-\xi)}$. Then, there exists a universal noise vector $\boldsymbol{v}$, such that $\|\boldsymbol{v}\|_2 \leq \rho$ and $\underset{x \sim \mu}{\mathbb{P}}\left(\hat{k}(\boldsymbol{x} + \boldsymbol{v}) \neq \hat{k}(\boldsymbol{x}) \text{ or } \hat{k}(\boldsymbol{x} - \boldsymbol{v}) \neq \hat{k}(\boldsymbol{x})\right) \geq 1 - \delta$.*

*Proof.* Define $\mathbb{S}$ to be the unit sphere centered at 0 in the subspace $\mathcal{S}$. Let $\rho = \frac{\sqrt{em}}{\delta(1-\xi)}$, and denote by $\rho\mathbb{S}$ the sphere scaled by $\rho$. We have

$$\underset{\boldsymbol{v}\sim\rho\mathbb{S}}{\mathbb{E}}\left(\underset{\boldsymbol{x}\sim\mu}{\mathbb{P}}\left(\hat{k}(\boldsymbol{x}+\boldsymbol{v})\neq\hat{k}(\boldsymbol{x})\text{ or }\hat{k}(\boldsymbol{x}-\boldsymbol{v})\neq\hat{k}(\boldsymbol{x})\right)\right)$$

$$=\underset{\boldsymbol{x}\sim\mu}{\mathbb{E}}\left(\underset{\boldsymbol{v}\sim\rho\mathbb{S}}{\mathbb{P}}\left(\hat{k}(\boldsymbol{x}+\boldsymbol{v})\neq\hat{k}(\boldsymbol{x})\text{ or }\hat{k}(\boldsymbol{x}-\boldsymbol{v})\neq\hat{k}(\boldsymbol{x})\right)\right)$$

$$\geq\underset{\boldsymbol{x}\sim\mu}{\mathbb{E}}\left(\underset{\boldsymbol{v}\sim\rho\mathbb{S}}{\mathbb{P}}\left(|\boldsymbol{r}(\boldsymbol{x})^T\boldsymbol{v}|-\|\boldsymbol{r}(\boldsymbol{x})\|_2^2\geq0\right)\right)$$

$$=\underset{\boldsymbol{x}\sim\mu}{\mathbb{E}}\left(\underset{\boldsymbol{v}\sim\rho\mathbb{S}}{\mathbb{P}}\left(|(P_{\mathcal{S}}\boldsymbol{r}(\boldsymbol{x})+P_{\mathcal{S}^{\text{orth}}}\boldsymbol{r}(\boldsymbol{x}))^T\boldsymbol{v}|-\|\boldsymbol{r}(\boldsymbol{x})\|_2^2\geq0\right)\right),$$

where $P_{\mathcal{S}^{\text{orth}}}$ denotes the projection operator on the orthogonal of $\mathcal{S}$. Observe that $(P_{\mathcal{S}^{\text{orth}}}\boldsymbol{r}(\boldsymbol{x}))^T\boldsymbol{v}=0$. Note moreover that $\|\boldsymbol{r}(\boldsymbol{x})\|_2^2=1$ by assumption. Hence, the above expression simplifies to

$$\underset{\boldsymbol{x}\sim\mu}{\mathbb{E}}\left(\underset{\boldsymbol{v}\sim\rho\mathbb{S}}{\mathbb{P}}\left(|(P_{\mathcal{S}}\boldsymbol{r}(\boldsymbol{x}))^T\boldsymbol{v}|-1\geq0\right)\right)$$

$$=\underset{\boldsymbol{x}\sim\mu}{\mathbb{E}}\left(\underset{\boldsymbol{v}\sim\mathbb{S}}{\mathbb{P}}\left(|(P_{\mathcal{S}}\boldsymbol{r}(\boldsymbol{x}))^T\boldsymbol{v}|\geq\rho^{-1}\right)\right)$$

$$\geq\underset{\boldsymbol{x}\sim\mu}{\mathbb{E}}\left(\underset{\boldsymbol{v}\sim\mathbb{S}}{\mathbb{P}}\left(\left|\frac{(P_{\mathcal{S}}\boldsymbol{r}(\boldsymbol{x}))^T}{\|P_{\mathcal{S}}\boldsymbol{r}(\boldsymbol{x})\|_2}\boldsymbol{v}\right|\geq\frac{\delta}{\sqrt{em}}\right)\right),$$

where we have used the assumption of the projection of $\boldsymbol{r}(\boldsymbol{x})$ on the subspace $\mathcal{S}$. Hence, it follows from Lemma 1 that

$$\underset{\boldsymbol{v}\sim\rho\mathbb{S}}{\mathbb{E}}\left(\underset{\boldsymbol{x}\sim\mu}{\mathbb{P}}\left(\hat{k}(\boldsymbol{x}+\boldsymbol{v})\neq\hat{k}(\boldsymbol{x})\text{ or }\hat{k}(\boldsymbol{x}-\boldsymbol{v})\neq\hat{k}(\boldsymbol{x})\right)\right)\geq1-\delta.$$

Hence, there exists a universal vector $\boldsymbol{v}$ of $\ell_2$ norm $\rho$ such that $\underset{\boldsymbol{x}\sim\mu}{\mathbb{P}}\left(\hat{k}(\boldsymbol{x}+\boldsymbol{v})\neq\hat{k}(\boldsymbol{x})\text{ or }\hat{k}(\boldsymbol{x}-\boldsymbol{v})\neq\hat{k}(\boldsymbol{x})\right)\geq1-\delta.$ □

## B  PROOF OF THEOREM 2

**Theorem 2.** *Let $\kappa>0,\delta>0$ and $m\in\mathbb{N}$. Assume that the quadratic decision boundary model $\mathscr{Q}(\boldsymbol{x},\rho)$ holds for almost all $\boldsymbol{x}\sim\mu$, with $\rho=\sqrt{\frac{2\log(2/\delta)}{m}}\kappa^{-1}+\kappa^{-1/2}$. Let $\mathcal{S}$ be a $m$ dimensional subspace such that*

$$\underset{\boldsymbol{v}\sim\mathbb{S}}{\mathbb{P}}\left(\forall\boldsymbol{u}\in\mathbb{R}^2,\alpha_x^{-1}\boldsymbol{u}^TH_z^{\boldsymbol{r}(\boldsymbol{x}),\boldsymbol{v}}\boldsymbol{u}\geq\kappa\|\boldsymbol{u}\|_2^2\right)\geq1-\beta\text{ for almost all }\boldsymbol{x}\sim\mu,$$

*where $H_z^{\boldsymbol{r}(\boldsymbol{x}),\boldsymbol{v}}=\Pi^TH_z\Pi$ with $\Pi$ an orthonormal basis of $\text{span}(\boldsymbol{r}(\boldsymbol{x}),\boldsymbol{v})$, and $\mathbb{S}$ denotes the unit sphere in $\mathcal{S}$. Then, there is a universal perturbation vector $\boldsymbol{v}$ such that $\|\boldsymbol{v}\|_2\leq\rho$ and $\underset{\boldsymbol{x}\sim\mu}{\mathbb{P}}\left(\hat{k}(\boldsymbol{x}+\boldsymbol{v})\neq\hat{k}(\boldsymbol{x})\right)\geq1-\delta-\beta.$*

*Proof.* Let $\boldsymbol{x}\sim\mu$. We have

$$\underset{\boldsymbol{v}\sim\rho\mathbb{S}}{\mathbb{E}}\left(\underset{\boldsymbol{x}\sim\mu}{\mathbb{P}}\left(\hat{k}(\boldsymbol{x}+\boldsymbol{v})\neq\hat{k}(\boldsymbol{x})\right)\right)$$

$$=\underset{\boldsymbol{x}\sim\mu}{\mathbb{E}}\left(\underset{\boldsymbol{v}\sim\rho\mathbb{S}}{\mathbb{P}}\left(\hat{k}(\boldsymbol{x}+\boldsymbol{v})\neq\hat{k}(\boldsymbol{x})\right)\right)$$

$$\geq\underset{\boldsymbol{x}\sim\mu}{\mathbb{E}}\left(\underset{\boldsymbol{v}\sim\rho\mathbb{S}}{\mathbb{P}}\left(\alpha_x^{-1}(\boldsymbol{v}-\boldsymbol{r})^TH_z(\boldsymbol{v}-\boldsymbol{r})+\boldsymbol{r}^T(\boldsymbol{v}-\boldsymbol{r})\geq0\right)\right)$$

$$=\underset{\boldsymbol{x}\sim\mu}{\mathbb{E}}\left(\underset{\boldsymbol{v}\sim\mathbb{S}}{\mathbb{P}}\left(\alpha_x^{-1}(\rho\boldsymbol{v}-\boldsymbol{r})^TH_z(\rho\boldsymbol{v}-\boldsymbol{r})+\boldsymbol{r}^T(\rho\boldsymbol{v}-\boldsymbol{r})\geq0\right)\right)$$

Using the assumptions of the theorem, we have

$$\mathbb{P}_{\boldsymbol{v} \sim \mathbb{S}} \left( \alpha_x^{-1} (\rho \boldsymbol{v} - \boldsymbol{r})^T H_z (\rho \boldsymbol{v} - \boldsymbol{r}) + \boldsymbol{r}^T (\rho \boldsymbol{v} - \boldsymbol{r}) \leq 0 \right)$$

$$\leq \mathbb{P}_{\boldsymbol{v} \sim \mathbb{S}} \left( \kappa \| \rho \boldsymbol{v} - \boldsymbol{r} \|_2^2 + \boldsymbol{r}^T (\rho \boldsymbol{v} - \boldsymbol{r}) \leq 0 \right) + \beta$$

$$\leq \mathbb{P}_{\boldsymbol{v} \sim \mathbb{S}} \left( \rho (1 - 2\kappa) \boldsymbol{v}^T \boldsymbol{r} + \kappa \rho^2 + (\kappa - 1) \leq 0 \right) + \beta$$

$$\leq \mathbb{P}_{\boldsymbol{v} \sim \mathbb{S}} \left( \rho (1 - 2\kappa) \boldsymbol{v}^T \boldsymbol{r} \leq -\epsilon \right) + \mathbb{P}_{\boldsymbol{v} \sim \mathbb{S}} \left( \kappa \rho^2 + (\kappa - 1) \leq \epsilon \right) + \beta,$$

for $\epsilon > 0$. The goal is therefore to find $\rho$ such that $\kappa \rho^2 + (\kappa - 1) \geq \epsilon$, together with $\mathbb{P}_{\boldsymbol{v} \sim \mathbb{S}} \left( \rho (1 - 2\kappa) \boldsymbol{v}^T \boldsymbol{r} \leq -\epsilon \right) \leq \delta$. Let $\rho^2 = \frac{\epsilon + 1}{\kappa}$. Using the concentration of measure on the sphere Matousek (2002), we have

$$\mathbb{P}_{\boldsymbol{v} \sim \mathbb{S}} \left( \boldsymbol{v}^T \boldsymbol{r} \leq \frac{-\epsilon}{\rho (1 - 2\kappa)} \right) \leq 2 \exp \left( -\frac{m \epsilon^2}{2 \rho^2 (1 - 2\kappa)^2} \right).$$

To bound the above probability by $\delta$, we set $\epsilon = C \frac{\rho}{\sqrt{m}}$, where $C = \sqrt{2 \log(2/\delta)}$. We therefore choose $\rho$ such that

$$\rho^2 = \kappa^{-1} \left( C \rho m^{-1/2} + 1 \right)$$

The solution of this second order equation gives

$$\rho = \frac{C \kappa^{-1} m^{-1/2} + \sqrt{\kappa^{-2} C^2 m^{-1} + 4 \kappa^{-1}}}{2} \leq C \kappa^{-1} m^{-1/2} + \kappa^{-1/2}.$$

Hence, for this choice of $\rho$, we have by construction

$$\mathbb{P}_{\boldsymbol{v} \sim \mathbb{S}} \left( \alpha_x^{-1} (\rho \boldsymbol{v} - \boldsymbol{r})^T H_z (\rho \boldsymbol{v} - \boldsymbol{r}) + \boldsymbol{r}^T (\rho \boldsymbol{v} - \boldsymbol{r}) \leq 0 \right) \leq \delta + \beta.$$

We therefore conclude that $\mathbb{E}_{\boldsymbol{v} \sim \rho \mathbb{S}} \left( \mathbb{P}_{\boldsymbol{x} \sim \mu} \left( \hat{k}(\boldsymbol{x} + \boldsymbol{v}) \neq \hat{k}(\boldsymbol{x}) \right) \right) \geq 1 - \delta - \beta$. This shows the existence of a universal noise vector $\boldsymbol{v} \sim \rho \mathbb{S}$ such that $\hat{k}(\boldsymbol{x} + \boldsymbol{v}) \neq \hat{k}(\boldsymbol{x})$ with probability larger than $1 - \delta - \beta$. □

## C    COMPLEMENTARY EXPERIMENTAL RESULTS

### C.1    EXPERIMENT IN FIG 7 (B)

We perform here similar experiment to Fig. 7 (b) on the VGG-16 and ResNet-18 architectures. It can be seen that, similarly to Fig. 7 (b), the proposed approach of generating universal perturbations through random sampling from the subspace $\mathcal{S}_c$ achieves high fooling rates (comparable to the algorithm in Moosavi-Dezfooli et al. (2017), and significantly higher than by using $\mathcal{S}_f$).

### C.2    TRANSFERABILITY OF UNIVERSAL PERTURBATIONS

Fig. 13 shows examples of normal cross-sections of the decision boundary across a *fixed* direction in $\mathcal{S}_c$, for the VGG-16 architecture (but where $\mathcal{S}_c$ is computed for *CaffeNet*). Note that the decision boundary across this *fixed* direction is positively curved for both networks, albeit computing this subspace for a distinct network. The sharing of $\mathcal{S}_c$ across different nets explains the transferability of universal perturbations observed in Moosavi-Dezfooli et al. (2017).

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

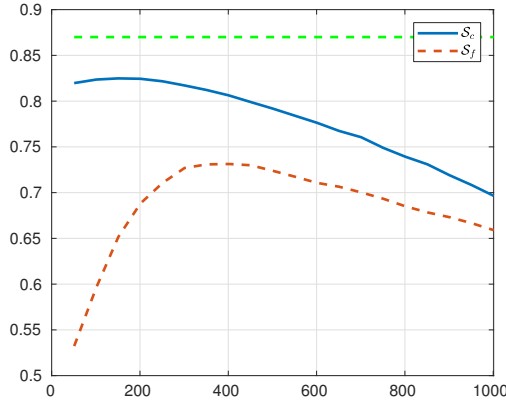

Figure 11: Same experiment as Fig. 7 (b) performed on VGG-16 architecture (CIFAR-10 dataset).

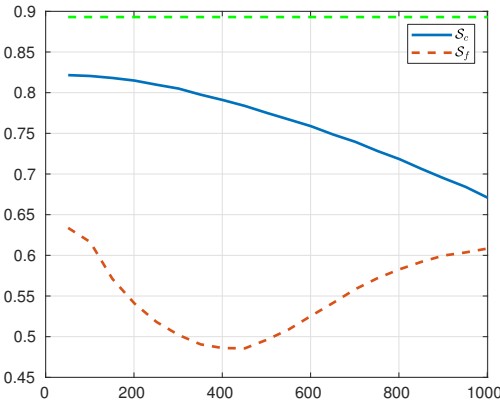

Figure 12: Same experiment as Fig. 7 (b) performed on ResNet-18 architecture (CIFAR-10 dataset).

Constantine Caramanis, Shie Mannor, and Huan Xu. Robust optimization in machine learning. In Suvrit Sra, Sebastian Nowozin, and Stephen J Wright (eds.), *Optimization for machine learning*, chapter 14. Mit Press, 2012.

Nicholas Carlini and David Wagner. Towards evaluating the robustness of neural networks. In *Security and Privacy (SP), 2017 IEEE Symposium on*, pp. 39–57. IEEE, 2017.

Moustapha Cisse, Piotr Bojanowski, Edouard Grave, Yann Dauphin, and Nicolas Usunier. Parseval networks: Improving robustness to adversarial examples. In *International Conference on Machine Learning (ICML)*, 2017.

Sanjoy Dasgupta and Anupam Gupta. An elementary proof of a theorem of johnson and lindenstrauss. *Random Structures & Algorithms*, 22(1):60–65, 2003.

Ivan Evtimov, Kevin Eykholt, Earlence Fernandes, Tadayoshi Kohno, Bo Li, Atul Prakash, Amir Rahmati, and Dawn Song. Robust physical-world attacks on machine learning models. *arXiv preprint arXiv:1707.08945*, 2017.

Alhussein Fawzi, Omar Fawzi, and Pascal Frossard. Analysis of classifiers' robustness to adversarial perturbations. *arXiv preprint arXiv:1502.02590*, 2015.

Alhussein Fawzi, Seyed Moosavi-Dezfooli, and Pascal Frossard. Robustness of classifiers: from adversarial to random noise. In *Neural Information Processing Systems (NIPS)*, 2016.

Ian J. Goodfellow, Jonathon Shlens, and Christian Szegedy. Explaining and harnessing adversarial examples. In *International Conference on Learning Representations (ICLR)*, 2015.

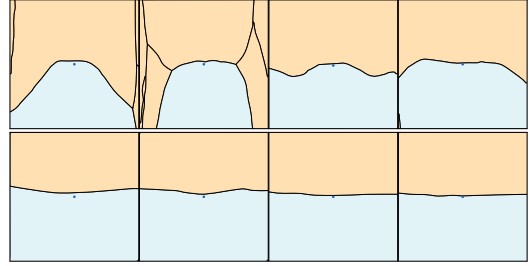

Figure 13: Transferability of the subspace $\mathcal{S}_c$ across different *networks*. The first row shows normal cross sections along a fixed direction in $\mathcal{S}_c$ for VGG-16, with a subspace $\mathcal{S}_c$ computed with CaffeNet. Note the positive curvature in most cases. To provide a baseline for comparison, the second row illustrates normal sections along random directions.

Shixiang Gu and Luca Rigazio. Towards deep neural network architectures robust to adversarial examples. *arXiv preprint arXiv:1412.5068*, 2014.

Kaiming He, Xiangyu Zhang, Shaoqing Ren, and Jian Sun. Deep residual learning for image recognition. In *IEEE Computer Vision and Pattern Recognition (CVPR)*, 2016.

Jan Hendrik Metzen, Mummadi Chaithanya Kumar, Thomas Brox, and Volker Fischer. Universal adversarial perturbations against semantic image segmentation. In *Proceedings of the IEEE Conference on Computer Vision and Pattern Recognition*, pp. 2755–2764, 2017.

Yangqing Jia, Evan Shelhamer, Jeff Donahue, Sergey Karayev, Jonathan Long, Ross Girshick, Sergio Guadarrama, and Trevor Darrell. Caffe: Convolutional architecture for fast feature embedding. In *ACM International Conference on Multimedia (MM)*, pp. 675–678, 2014.

Valentin Khrulkov and Ivan Oseledets. Art of singular vectors and universal adversarial perturbations. *arXiv preprint arXiv:1709.03582*, 2017.

Alex Krizhevsky, Ilya Sutskever, and Geoffrey Hinton. Imagenet classification with deep convolutional neural networks. In *Advances in Neural Information Processing Systems (NIPS)*, pp. 1106–1114, 2012.

Gert Lanckriet, Laurent Ghaoui, Chiranjib Bhattacharyya, and Michael Jordan. A robust minimax approach to classification. *The Journal of Machine Learning Research*, 3:555–582, 2003.

Y. LeCun, L. Bottou, Y. Bengio, and P. Haffner. Gradient-based learning applied to document recognition. *Proceedings of the IEEE*, 86(11):2278–2324, 1998.

Jeffrey M Lee. *Manifolds and differential geometry*, volume 107. American Mathematical Society Providence, 2009.

Min Lin, Qiang Chen, and Shuicheng Yan. Network in network. In *International Conference on Learning Representations (ICLR)*, 2014.

Aleksander Madry, Aleksandar Makelov, Ludwig Schmidt, Dimitris Tsipras, and Adrian Vladu. Towards deep learning models resistant to adversarial attacks. *arXiv preprint arXiv:1706.06083*, 2017.

Jiri Matousek. *Lectures on discrete geometry*, volume 108. Springer New York, 2002.

Guido F Montufar, Razvan Pascanu, Kyunghyun Cho, and Yoshua Bengio. On the number of linear regions of deep neural networks. In *Advances In Neural Information Processing Systems*, pp. 2924–2932, 2014.

Seyed-Mohsen Moosavi-Dezfooli, Alhussein Fawzi, and Pascal Frossard. Deepfool: a simple and accurate method to fool deep neural networks. In *IEEE Conference on Computer Vision and Pattern Recognition (CVPR)*, 2016.

Seyed-Mohsen Moosavi-Dezfooli, Alhussein Fawzi, Omar Fawzi, and Pascal Frossard. Universal adversarial perturbations. In *IEEE Conference on Computer Vision and Pattern Recognition (CVPR)*, 2017.

Konda Reddy Mopuri, Utsav Garg, and R Venkatesh Babu. Fast feature fool: A data independent approach to universal adversarial perturbations. In *British Machine Vision Conference (BMVC)*, 2017.

Nicolas Papernot, Patrick McDaniel, Xi Wu, Somesh Jha, and Ananthram Swami. Distillation as a defense to adversarial perturbations against deep neural networks. *arXiv preprint arXiv:1511.04508*, 2015.

Ben Poole, Subhaneil Lahiri, Maithreyi Raghu, Jascha Sohl-Dickstein, and Surya Ganguli. Exponential expressivity in deep neural networks through transient chaos. In *Advances In Neural Information Processing Systems*, pp. 3360–3368, 2016.

Mahmood Sharif, Sruti Bhagavatula, Lujo Bauer, and Michael K Reiter. Accessorize to a crime: Real and stealthy attacks on state-of-the-art face recognition. In *Proceedings of the 2016 ACM SIGSAC Conference on Computer and Communications Security*, pp. 1528–1540. ACM, 2016.

Christian Szegedy, Wojciech Zaremba, Ilya Sutskever, Joan Bruna, Dumitru Erhan, Ian Goodfellow, and Rob Fergus. Intriguing properties of neural networks. In *International Conference on Learning Representations (ICLR)*, 2014.

Pedro Tabacof and Eduardo Valle. Exploring the space of adversarial images. *IEEE International Joint Conference on Neural Networks*, 2016.

Thomas Tanay and Lewis Griffin. A boundary tilting persepective on the phenomenon of adversarial examples. *arXiv preprint arXiv:1608.07690*, 2016.

David Warde-Farley, Ian Goodfellow, T Hazan, G Papandreou, and D Tarlow. Adversarial perturbations of deep neural networks. *Perturbations, Optimization, and Statistics*, 2016.

Huan Xu, Constantine Caramanis, and Shie Mannor. Robustness and regularization of support vector machines. *The Journal of Machine Learning Research*, 10:1485–1510, 2009.

