# OpenReview forum: "Robustness of Classifiers to Universal Perturbations: A Geometric Perspective"
_ICLR.cc/2018/Conference — Accept (Poster)_

### Official Review · AnonReviewer3 · 2017-11-27
**The paper provides an interesting analysis linking the geometry of classifier decision boundaries to  small universal adversarial perturbations.**

**Rating:** 6
**Confidence:** 4

**Review:**

The paper is written well and clear.   The core contribution of the paper is the illustration that: under the assumption of flat, or curved decision boundaries with positive curvature small universal adversarial perturbations exist.

Pros: the intuition and geometry is rather clearly presented.

Cons:
References to "CaffeNet"  and "LeNet" (even though the latter is well-known) are missing.  In the experimental section used to validate the main hypothesis that the deep networks have positive curvature decision boundaries, there is no description of how these networks were trained.

It is not clear why the authors have decided to use out-dated 5-layer "LeNet"  and NiN (Network in network) architectures instead of more recent and much better performing architectures (and less complex than NiN architectures). It would be nice to see how the behavior and boundaries look in these cases.

The conclusion is speculative:
"Our analysis hence shows that to construct classifiers that are robust to universal perturbations, it
is key to suppress this subspace of shared positive directions, which can possibly be done through
regularization of the objective function. This will be the subject of future works."

It is clear that regularization should play a significant role in shaping the decision boundaries. Unfortunately, the paper does not provide details at the basic level, which algorithms,  architectures, hyper-parameters or regularization terms are used. All these factors should play a very significant role in the experimental validation of their hypothesis.

Notes: I did not check the proofs of the theorems in detail.

---

> ### Author Response · Authors · 2017-12-13
> **Reply to reviewer**
>
> We thank the reviewer for the comments that helped improve the manuscript. Please see clarifications below.
>
> 1. We have updated the manuscript with references for the networks, as well as a description of how these networks were trained as requested by the reviewer.
>
> 2. To address the Reviewer concern, we have conducted new experiments on more architectures, in particular ResNet-18, VGG-16 for CIFAR-10 and ResNet-152 for ImageNet; all confirm and validate our results. Specifically,
>
> * Fig. 5 and 6 were updated with decision boundaries of ResNet-18 for CIFAR-10 and ResNet-152 for ImageNet.
> * We have conducted the same experiment as in Fig. 7 (b) for VGG-16 and ResNet-18 architectures. Please see Appendix C for the figures.
> * As also requested by Reviewer 1, we have shown visual examples on ImageNet of the universal perturbations computed using the curvature-based proposed approach. Please see Fig. 8.
>
> The new experiments confirm that our conclusions hold equally well on modern architectures; in particular, these new results confirm that the existence of universal perturbations is due to the existence of shared positively curved directions in the decision boundary of deep networks.
>
> 3. While our conclusion is indeed speculative, we believe that our analysis (in particular the fact that universal perturbations are random vectors in subspace S_c) can be leveraged to improve the robustness to universal perturbations. Other authors have actually already used our analysis to counter universal perturbations in a very recent paper [Anonymous, 2017]*. The authors specifically eliminate universal perturbations through random sampling from this subspace, and training a "denoising" module to effectively project on the orthogonal of this subspace. This is indeed a very simple way of using the proposed analysis, and we believe that such analysis will lead to more ways to counter universal perturbations.
>
> *: We anonymized this paper, as it is citing a technical report of ours and might violate the double blind policy.

---

### Official Review · AnonReviewer1 · 2017-11-28
**The main issue I am having is what are the applicable insight from the analysis**

**Rating:** 5
**Confidence:** 3

**Review:**

This paper discusses universal perturbations - perturbations that can mislead a trained classifier if added to most of input data points. The main results are two fold: if the decision boundary are flat (such as linear classifiers), then the classifiers tend to be vulnerable to universal perturbations when the decision boundaries are correlated. If the decision boundary are curved, then vulnerability to universal perturbations is directly resulted from existence of shared direction along with the decision boundary positively curved. The authors also conducted experiments to show that deep nets produces decision boundary that satisfies the curved model.

The main issue I am having is what are the applicable insight from the analysis:

1. Why is universal perturbation an important topic (as opposed to adversarial perturbation).
2. Does the result implies that we should make the decision boundary more flat, or curved but on different directions? And how to achieve that? It might be my mis-understanding but from my reading a prescriptive procedure for universal perturbation seems not attained from the results presented.

---

> ### Author Response · Authors · 2017-12-13
> **Reply to reviewer**
>
> We thank the reviewer for the comments. Please see clarifications below.
>
> 1. Universal perturbations are static images that can be used by adversaries to fool a classifier (no need to run an optimization procedure to fool each new image); classifiers hence need to be robust to this excessively simple perturbation model. Adversarial perturbations are image-specific and do not generalize well across different images.
> The existence of universal perturbations is also informative for the geometry of the classification boundaries, which is one step towards better understanding the fundamental properties of deep networks.
>
> 2. The goal of the paper is not (yet) to improve the design of classifiers, but to gain insight through their analysis. It is beyond the scope of a single paper to prescribe procedures to improve robustness by modifying the curvature of classification regions.
> Nevertheless, we should mention that our analysis (in particular, the fact that universal perturbations are random vectors in subspace S_c) has already been used by others to provide a constructive procedure to combat universal perturbations [Anonymous, 2017]*. The authors specifically eliminate universal perturbations through random sampling from this subspace, and training a "denoising" module to effectively project on the orthogonal of this subspace.
> This is indeed a very simple way of using the proposed analysis, and we believe that such analysis will inspire more ways to counter universal perturbations.
>
> *: We anonymized this paper, as it is citing a technical report of ours and might violate the double blind policy.

---

### Official Review · AnonReviewer2 · 2017-12-06
**Universal perturbations as a consequence of positive curvature**

**Rating:** 7
**Confidence:** 3

**Review:**

The paper develops models which attempt to explain the existence of universal perturbations which fool neural networks — i.e., the existence of a single perturbation which causes a network to misclassify most inputs. The paper develops two models for the decision boundary:

(a) A locally flat model in which the decision boundary is modeled with a hyperplane and the normals two the hyperplanes are assumed to lie near a low-dimensional linear subspace.

(b) A locally positively curved model, in which there is a positively curved outer bound for the collection of points which are assigned a given label.

The paper works out a probabilistic analysis arguing that when either of these conditions obtains, there exists a fooling perturbation which affects most of the data.

The theoretical analysis in the paper is straightforward, in some sense following from the definition. The contribution of the paper is to posit these two conditions which can predict the existence of universal fooling perturbations, argue experimentally that they occur in (some) neural networks of practical interest.

One challenge in assessing the experimental claims is that practical neural networks are nonsmooth; the quadratic model developed from the hessian is only valid very locally. This can be seen in some of the illustrative examples in Figure 5: there *is* a coarse-scale positive curvature, but this would not necessarily come through in a quadratic model fit using the hessian. The best experimental evidence for the authors’ perspective seems to be the fact that random perturbations from S_c misclassify more points than random perturbations constructed with the previous method.

I find the topic of universal perturbations interesting, because it potentially tells us something structural (class-independent) about the decision boundaries constructed by artificial neural networks. To my knowledge, the explanation of universal perturbations in terms of positive curvature is novel. The paper would be much stronger if it provided an explanation of *why* there exists this common subspace of universal fooling perturbations, or even what it means geometrically that positive curvature obtains at every data point.

Visually, these perturbations seem to have strong, oriented local high-frequency content — perhaps they cause very large responses in specific filters in the lower layers of a network, and conventional architectures are not robust to this?

It would also be nice to see some visual representations of images perturbed with the new perturbations, to confirm that they remain visually similar to the original images.

---

> ### Author Response · Authors · 2017-12-13
> **Reply to reviewer**
>
> We thank the reviewer for the comments. Please see clarifications below.
>
> - As requested, we have added visual representations of images perturbed with the new perturbation (see Fig. 8).
>
> - We agree curvature is indeed only informative of the local structure of the decision boundary, but a first step to understand it. In the experiments, we have looked at coarse scale second-order information through a finite difference of gradients. This is indeed inevitable as state-of-the art networks using ReLU have theoretically vanishing Hessian almost everywhere.
>
> - Visual appearance of universal perturbations: That is an interesting question. Our focus in this paper was more oriented towards explaining the existence of universal perturbations through an investigation of the geometry of the decision boundary. Interpreting the visual appearance of universal perturbations requires to draw a link between the weights in the lower layers with the curvature of the decision boundary. This would definitely be a fascinating connection, that we would like to work on in the future.

---

### Decision · Program_Chairs · 2018-01-29
**ICLR 2018 Conference Acceptance Decision**

**Decision:**

Accept (Poster)

**Comment:**

The idea of universal perturbation is definitely interesting and well carried out in that paper.